# RuvC uses dynamic probing of the Holliday junction to achieve sequence specificity and efficient resolution

Karolina Maria Górecka [1,5], Miroslav Krepl [2,5], Aleksandra Szlachcic [1], Jarosław Poznański [3], Jiří Šponer [2,4] & Marcin Nowotny [1]

Holliday junctions (HJs) are four-way DNA structures that occur in DNA repair by homologous recombination. Specialized nucleases, termed resolvases, remove (i.e., resolve) HJs. The bacterial protein RuvC is a canonical resolvase that introduces two symmetric cuts into the HJ. For complete resolution of the HJ, the two cuts need to be tightly coordinated. They are also specific for cognate DNA sequences. Using a combination of structural biology, biochemistry, and a computational approach, here we show that correct positioning of the substrate for cleavage requires conformational changes within the bound DNA. These changes involve rare high-energy states with protein-assisted base flipping that are readily accessible for the cognate DNA sequence but not for non-cognate sequences. These conformational changes and the relief of protein-induced structural tension of the DNA facilitate coordination between the two cuts. The unique DNA cleavage mechanism of RuvC demonstrates the importance of high-energy conformational states in nucleic acid readouts.

[1] Laboratory of Protein Structure, International Institute of Molecular and Cell Biology, 4 Trojdena St., 02-109 Warsaw, Poland. [2] Institute of Biophysics of the Czech Academy of Sciences, Kralovopolska 135, 612 65 Brno, Czech Republic. [3] Institute of Biochemistry and Biophysics Polish Academy of Sciences, 5a Pawinskiego St., 02-106 Warsaw, Poland. [4] Regional Centre of Advanced Technologies and Materials, Faculty of Science, Palacky University Olomouc, Slechtitelu 27, 771 46 Olomouc, Czech Republic. [5] These authors contributed equally: Karolina Maria Górecka, Miroslav Krepl. Correspondence and requests for materials should be addressed to M.K. (email: krepl@ibp.cz) or to M.N. (email: mnowotny@iimcb.gov.pl)

Holliday junctions (HJs) are DNA structures in which two duplexes are joined by the exchange of strands. Holliday junctions are involved in various pathways of genetic information processing, notably homologous recombination (HR)[1], in which dangerous double-stranded DNA breaks are repaired. During HR repair, an identical or nearly identical (homologous) stretch of undamaged DNA is used to repair the damaged part. This process requires that both damaged and undamaged DNA fragments are joined into the HJ[2]. In higher organisms, HR is also involved in meiosis that generates genetic diversity by reshuffling genes.

Once their function is complete, HJs need to be efficiently removed because the permanent joining of chromosomes can lead to severe genetic instability. The removal of HJs can occur through actions of specialized nucleases, termed resolvases. Resolvases are present in all forms of life[3]. The model enzyme from this group is the bacterial resolvase RuvC. It belongs to the retroviral integrase superfamily that has a characteristic RNase H fold and catalytic mechanism that involves two divalent metal ions[4]. RuvC is a dimeric enzyme that resolves HJs by introducing two symmetric 5′-phosphorylated cuts near the center of the HJ (i.e., the exchange point)[5–10]. The cuts occur at the 5′-A/TTT↓G/C-3′ consensus sequence[5,11–13], which has important biological functions. The sequence-selectivity of cleavage restricts productive resolution to homologous sequences, in which the preferred sequence is presented to both subunits of the enzyme. One salient feature of RuvC is that it performs complete HJ resolution, which requires coordination between the two cuts. The second cut must occur before the substrate with a single cut can dissociate from RuvC. This process was proposed to occur through a so-called nick-counternick mechanism, in which the first cut greatly accelerates the second cut[14–16], but the structural basis of this mechanism was not clarified.

The crystal structures of free and complexed RuvC were first reported in 1994 and 2013, respectively[17,18]. The complex structure showed that the HJ bound in a tetrahedral conformation with two of its phosphates located near the two active sites of the RuvC dimer, each 1 nucleotide (nt) from the exchange point toward the 3′ end of the cleaved strand. Interestingly, although cleavage occurs only at the consensus sequence, DNA binding by RuvC occurs in a sequence-independent manner[5,12,19]. In the RuvC–HJ complex structure that was reported in 2013, the bound DNA substrate has a sequence that partially matches RuvC's consensus near the active sites, but no potential sequence-specific protein–DNA contacts were observed. Thus, the structural basis of the enzyme's sequence specificity was unclear.

The indiscriminate binding of HJ sequences by RuvC contrasts with its sequence-specific requirements for catalysis, implying that events occurring after complex formation are responsible for the enzyme's sequence specificity. The lack of direct sequence-specific contacts suggested that RuvC's consensus sequence readout could involve dynamic sampling of the HJ substrate and/or high-energy states of the substrate, both of which are difficult to capture in the crystal structures. Thus, we had to apply other experimental approaches to resolve these issues.

We first solved a new, higher resolution structure of the RuvC–HJ complex that was suitable for extensive molecular dynamics simulations on a microsecond timescale. The simulation results were then verified by biochemical experiments. We learned that both the nick-counternick mechanism and RuvC's sequence preference can be explained by the fact that the DNA substrate needs to undergo specific conformational rearrangements before the cuts can occur. Based on our data, we propose a comprehensive model of the action of RuvC on the HJ substrate. Our findings underscore the importance of conformational probing of DNA and high-energy transient states in nucleic acid recognition.

## Results

### RuvC–HJ structure reveals tension at the DNA exchange point.
We previously reported a crystal structure of the *Thermus thermophilus* (*Tt*) RuvC–HJ complex that was solved at 3.8 Å resolution[18]. To obtain more detailed insights into substrate recognition by the enzyme, we performed new X-ray diffraction experiments that are described in the Methods section and resulted in better resolution of 3.4 Å (Supplementary Table 1). The new structure was refined to a low $R_{free}$ of 23.3% (90th percentile of structures of similar resolution), had a good fit between the model and electron density maps and the Molprobity score in the 100th percentile (for more details see "Methods"). In the new structure, the electron density maps were better defined than in the previous one (sample electron density maps are shown in Supplementary Fig. 1). This was especially important in the key region of protein that protrudes into the opening at the exchange point of the HJ, forming an element we term "wedge" (Fig. 1). Many of the side chains from the wedge directly stack with DNA base pairs that are closest to the junction point. The higher resolution X-ray diffraction data revealed that they cause deformations of these base pairs (Fig. 1c). Specifically, they buckle the base pairs by 17.1, 8.9, 34.9, and 33.8 degrees for DNA arms A1, A2, B1, and B2, respectively (A1 and A2 are the cleaved arms; Fig. 1b).

Further analysis of the new 3.4 Å structure also indicated that the RuvC–HJ complex did not adopt a fully catalytic configuration. Namely, the distance between the phosphorus atoms of the scissile phosphates and C-α atoms of the catalytic Asp7 residues was greater than 8 Å, whereas in the structures of related retroviral integrase superfamily enzymes, it is within the range of 7.25–7.5 Å (PDB ID: 1ZBI[20], 3O3G[21], and 4E7I[22]). This implies that structural rearrangement of the complex must occur to achieve the catalytically active configuration. We noted that although RuvC binds HJs with high affinity, its enzymatic activity is relatively low[18], which could indicate that its catalytic geometry consists of a rarely (transiently) populated state.

### Simulations of RuvC–DNA complex sample catalytic geometries.
The new 3.4 Å crystal structure allowed us to perform MD simulations to obtain further insights into the catalytic mechanism of RuvC and its substrate specificity. The initial simulation of the X-ray structure showed a stable and symmetric protein–DNA interface, details of which are described in the Supporting Information (Supplementary Fig. 2c). The stable behavior of the structural model confirmed its sufficient quality and suitability for MD analysis. In the new RuvC–HJ X-ray structure, the DNA sequence that is located near the active sites only partially matched the cleavage consensus, and the complex did not correspond to the catalytic configuration. Nevertheless, the subsequent MD simulations in which we replaced the non-cognate DNA sequences with cognate DNA sequences showed that a catalytic-like geometry was spontaneously visited ~2% of the simulation time, without any additional restraints. Note that we define "catalytic-like" as situations in which the majority but not all of the geometrical conditions of the catalysis are met, including $Mg^{2+}$ coordination by individual amino acids and phosphate oxygen and the presence of a water molecule that is positioned for an in-line attack. Additional comments on the active sites and their metal ions can be found in the Supplementary Notes.

### Catalytic geometry requires structural changes in the DNA.
To explore catalytic action of the RuvC–DNA complex, we performed several simulations with a set of distance restraints that were designed to promote the expected catalytic geometry at both catalytic sites (see Methods, Supplementary Figs. 2a, b and 3). The

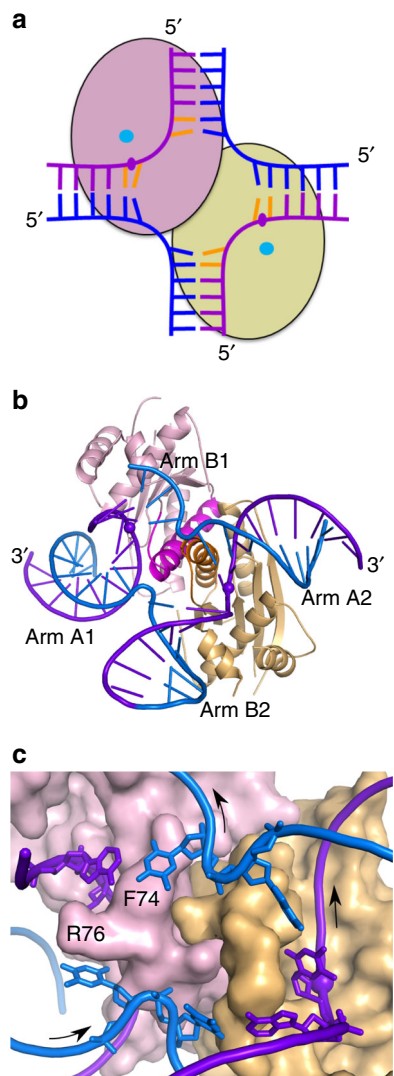

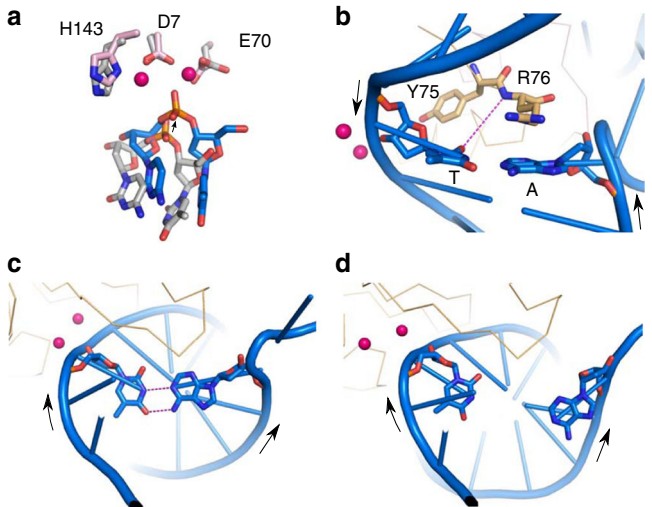

**Fig. 1** Overall structure of *Tt*-RuvC–DNA complex and deformations of base pairing around the exchange point of the HJ. **a** Scheme of the complex structure. The subunits of the RuvC dimer are shown in pink and yellow green. The DNA is shown in blue (non-cleaved strands) and purple (cleaved strands) ladder-like representation. The scissile phosphates are shown as purple ovals, and the active sites are shown as cyan circles. The nucleotides of the consensus sequence are in orange. **b** Crystal structure of RuvC–HJ complex. The arms of the HJ are labeled, the scissile phosphates are indicated as spheres, and protein α-helices B with preceding loops that form the wedge element are shown in a darker color. Please note that both arm B1 and B2 are terminated with loops comprising three thymine residues, but in arm B1 loop is not visible in electron density maps. **c** Close-up view of the exchange point of the HJ. The protein is shown in surface representation, and the side chains of selected amino acids of the wedge element are labeled. Base pairs around the exchange point are shown as sticks. Arrows show the 5′ to 3′ polarity of DNA strands

**Fig. 2** Structural changes in MD simulations of the RuvC–DNA complex associated with the formation of catalytic-like geometry. **a** Close-up view of the active site of RuvC. Active-site residues and bases on both sides of the scissile phosphate are shown as blue and gray sticks for the MD simulations model and X-ray structure, respectively. Magnesium ions are shown as pink spheres. The black arrow indicates the motion direction of the scissile phosphate. **b** A new H-bond interaction between the protein backbone and scissile thymine formed in the simulations of the RuvC–DNA complex (purple). Helices B and preceding loops are shown in wire representation. Arrows show 5′ to 3′ polarity of DNA strands. **c**, **d** Pairing of the scissile T–A base pair. Arrows show 5′ to 3′ polarity of DNA strands. **c** A stable base pair in the crystal structure. **d** The broken base pair in MD simulations

the dynamics of the complex that possesses a catalytically relevant geometry because it is not expected to occur frequently within the simulation timescale. In fact, the relatively low activity of *Tt*-RuvC could suggest that catalytic geometry is rarely populated.

All of the simulations showed that the formation of catalytic geometry was associated with structural changes within the RuvC–DNA complex. The most important of these occurred within the DNA substrate. We observed a shift of the DNA backbone around the scissile phosphate which brought it closer to the magnesium cofactors and catalytic residues of the protein (Fig. 2a). As expected, this movement strained the DNA substrate, particularly the T–A base pair on the 5′ side of the scissile phosphate (hereinafter referred to as the "scissile base pair"). Interestingly, RuvC appeared to compensate for this by establishing new protein–DNA interactions with the distorted base pair. In the majority of cases, a new H-bond formed between the O2 base atom of the scissile thymidine and the backbone amide of either Arg76 or, less frequently, Tyr75 (Fig. 2b). In our simulations, we occasionally observed the complete loss of base pairing of the scissile T–A base pair (Fig. 2c, d). This occurred more often in simulations in which the catalytic geometry was promoted by the distance restraints (Supplementary Table 2). We posit that the base pairing interactions of the scissile T–A base pair hamper the movement of the scissile phosphate toward the active site. Therefore, breaking the base pair could be required to enable the catalytic configuration.

**Arg76 side-chain promotes conformational changes in the DNA.** Molecular dynamics simulations further showed that the Arg76 side-chain of RuvC participated in the observed destabilization of the scissile T–A base pair (Fig. 3, Supplementary Movie 1). The Arg76 side-chain often formed either stacking or H-bond

use of the restraints to impose geometry that is conducive to catalysis can be justified by the following. First, the force-field description may be imbalanced against the geometry that is conducive to catalysis. Second, catalysis may proceed via a rarely populated but highly reactive conformation that is different from the dominantly sampled ground-state conformation of the RuvC–DNA complex in solution. The latter scenario appears to be common in systems that catalyze nucleic acid backbone cleavage[23]. Therefore, a set of distance restraints was used to explore

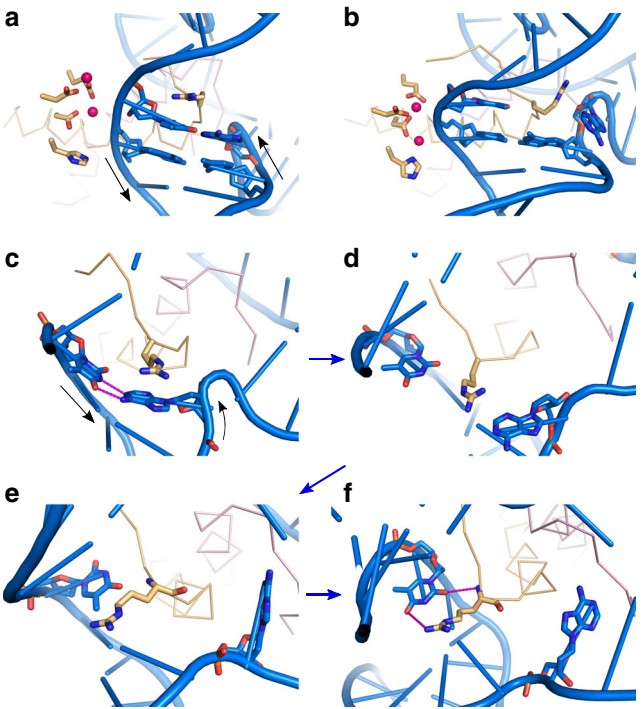

**Fig. 3** Examples of Arg76 conformations observed in MD simulations. The structure before (**a**) and after (**b**) the catalytic-like geometry is established. The base pairs on both sides of the scissile phosphate and Arg76 side-chain are shown as sticks, and the nearby backbone of helices B and preceding loops is shown as a wire representation. Arrows show 5′ to 3′ polarity of DNA strands. **c** Stacking between Arg76 and adenine base. Black arrows show 5′ to 3′ polarity of DNA strands. **d** Disruption of the base pair. **e** Flipping out of the adenine base. **f** Interactions that form with the scissile thymine. The blue arrows indicate the sequence of events. The dashed purple lines indicate H-bonds

interactions with the bases of the disrupted T–A base pair. It also frequently directly displaced the adenine base and was involved in the majority of the observed base pair disruptions (Fig. 3). An important observation in our simulations was that scissile T–A base pair disruption often involved flipping of the adenine base to the solvent (Fig. 3d, e). This base pair distortion sometimes occurred spontaneously, even without the involvement of Arg76, indicating that the process can proceed via multiple pathways.

Thus, the simulations suggested a dual role for the Arg76 side-chain. First, it functions as a structural probe by which RuvC protein can interact with and ultimately disrupt the scissile T–A base pair (Fig. 3). Second, after disrupting the scissile base pair, the Arg76 side-chain is able to form interactions with the unpaired thymine or adenine. This effectively prevents a rapid re-approach of the bases and re-formation of the T–A base pair (Fig. 3). Arg76 could influence the speed of the enzymatic reaction by both inducing instability in the scissile T–A base pair and prolonging the lifetime of the disrupted states that are then conducive to catalysis. Further prolongation of the disrupted states may also derive from specific dynamics of the displaced adenine base. In our MD simulations, it randomly fluctuated while exposed to the solvent. However, enhanced-sampling REST2 simulations revealed that adenines from both scissile base pairs eventually formed a very stable stacking interaction across the junction that was further aided by Arg76 (Supplementary Fig. 4). This process could be an important component of the overall free-energy landscape that leads to catalysis and is described in detail in the Supporting Notes.

**Biochemical experiments confirm the dynamics of the DNA.** The simulations of the RuvC–DNA complex suggested that formation of the catalytic interaction may be accompanied by breaking of the scissile T–A base pair with simultaneous flipping of the adenine base away from the helical structure of the DNA. To experimentally verify that this flipping occurs, we used HJ substrates with a 2-aminopurine substitution of selected adenines. 2-Aminopurine can base pair with thymine similarly to adenine[24], and its fluorescence significantly increases upon its removal from the duplex structure[25]. Thus, it can be used to probe conformational changes in A–T/T–A base pairs of double-stranded nucleic acids. We prepared HJ substrates with individual substitutions of adenine with 2-aminopurine in two positions. In the first substrate, the 2-aminopurine was located opposite the thymine on the 5′ side of the scissile phosphate (substrate AP1). In the second substrate, the 2-aminopurine was located 3 bp from the exchange point (AP2; Fig. 4a). Our assumption was that upon binding by RuvC, the 2-aminopurine in AP1 could be flipped out as suggested by the simulations, whereas it should remain in the duplex in AP2. Indeed, upon the addition of RuvC, fluorescence increased more than eightfold for AP1, whereas it did not change for AP2 (Fig. 4b). This confirmed the results of our MD simulations that showed that adenine of the scissile T–A base pair could be flipped out.

We next assessed the involvement of the Arg76 side-chain in adenine flipping. When the *Tt*-RuvC R76A mutant was used in combination with AP1, the change in fluorescence was approximately fivefold, which is less than for wild-type protein. As expected, no change was observed for the AP2 substrate (Fig. 4b). This is consistent with the simulations that showed that adenine flipping could occur spontaneously but was promoted by the side-chain of Arg76.

**Removal of the scissile base pair increases activity.** The MD simulations also suggested that breaking of the scissile T–A base pair could be associated with catalytic geometry. We hypothesized that replacing adenine with an abasic site would alleviate the need to break base pairing and could increase enzymatic activity. To verify this, we prepared three variants of the HJ substrate that contained abasic sites (HJ-Ab). The first variant contained an abasic site that was opposite to the thymine on the 5′ side of the scissile phosphate (HJ-Ab1A). The second variant had the abasic site in the same position but at the other catalytic site (HJ-Ab1B). The third variant contained both abasic sites (HJ-Ab2; Fig. 4c). As controls, we used unmodified substrate (HJ-C) and substrates with abasic site located 4 nucleotides upstream (HJ-Ab1A_-4) or 5 nucleotides downstream (HJ-AB1A_5) from the position of the flipped out adenine.

We first verified the affinity of *Tt*-RuvC for the selected substrates using fluorescence anisotropy. Both the HJ-C and HJ-Ab substrates bound with similar affinity, suggesting that the introduction of abasic sites did not affect the binding of RuvC to the HJ (Supplementary Table 3). We then performed a resolvase assay using fluorescently labeled substrates. For HJ-C, 50% cleavage by RuvC was observed after 120 min (Fig. 4d, Supplementary Fig. 5). The reaction was markedly more efficient for all three HJ-Ab substrates, with 65–80% cleavage after 120 min (Fig. 4d; Supplementary Fig. 5). We also tested the R76A RuvC mutant in our experiments, which had very little activity on the HJ-C (~20% cleavage after 120 min). However, when it was mixed with HJ-Ab substrates, its activity was essentially rescued, and 60% cleavage was observed after 120 min (Fig. 4e, Supplementary Fig. 5). This effect was specific. The enhancement of the activity of *Tt*-RuvC was not observed when the abasic site was introduced in other locations in the non-cleaved strand in the

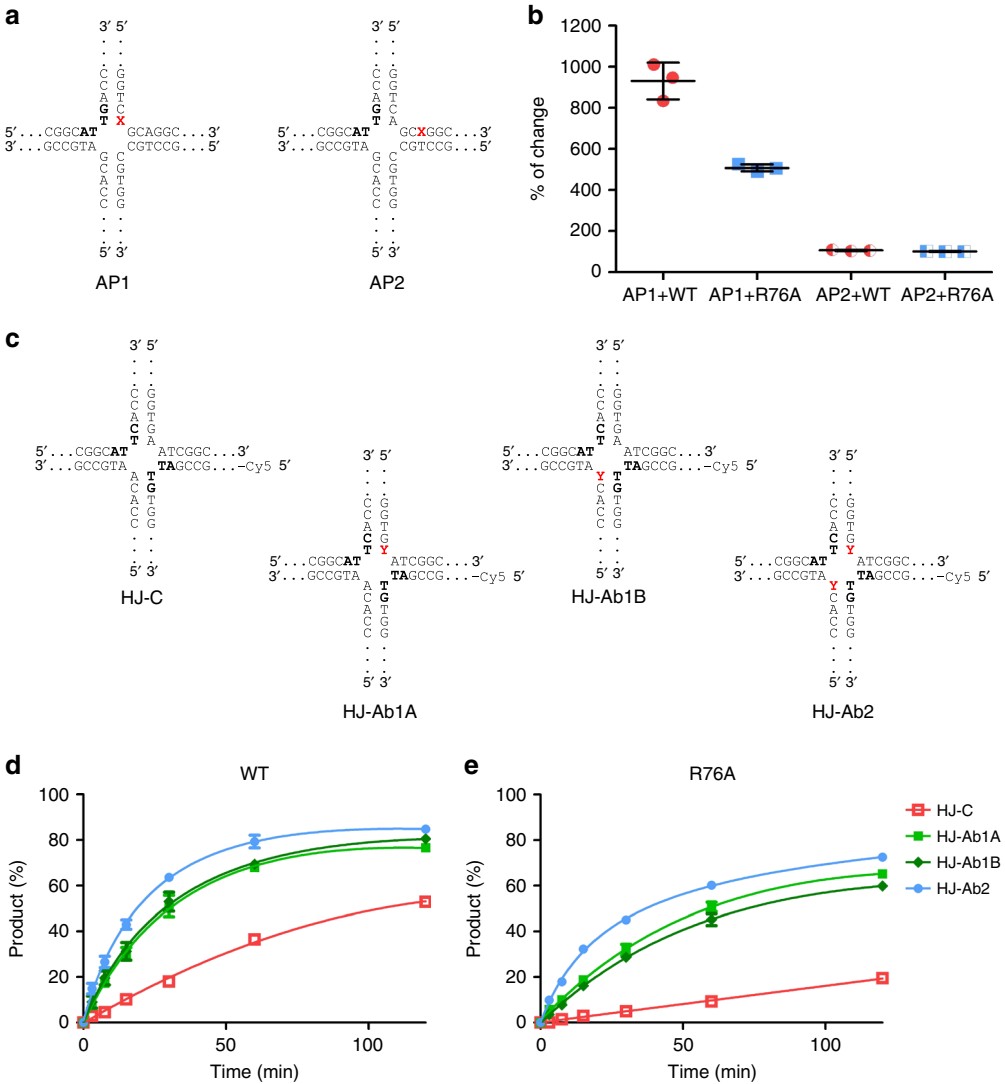

**Fig. 4** Role of Arg76-mediated base pair disruption and base flipping in *Tt*-RuvC activity. **a** Schemes of the HJ substrates that were used in measurements of the fluorescence of 2-aminopurine HJ upon binding to *Tt*-RuvC. The red X indicates 2-aminopurine. **b** Change in fluorescence after mixing the 2-aminopurine substrates with different *Tt*-RuvC variants. The results are expressed as the percent change in DNA fluorescence upon the addition of protein. Data from three independent experiments were averaged and plotted for each value. The error bars represent the standard deviation. Dot plots are showing individual data points. **c** Scheme of HJs that were used to measure the activity of *Tt*-RuvC (wild-type and R76A) on substrates with abasic sites. The red Y indicates an abasic nucleotide. **d**, **e** Resolving activity of the *Tt*-RuvC variants [wild-type (**d**) and R76A (**e**)] that acted on the control and abasic site substrates. Data from four independent experiments were averaged and plotted for each timepoint. Error bars represent the standard deviation

HJ substrates HJ-Ab1A_-4 and HJ-Ab1A_5 (Supplementary Fig. 6). To further verify these results, we also performed additional experiments for wild-type protein and R76A variant with HJ-Ab2 substrate. We used a single timepoint (30 min) and different substrate:protein ratios (1:0.25 to 1:10) (Supplementary Fig. 7). For all tested ratios the results were in agreement with the outcome of the initial time-course experiment. In conclusion, we found that an abasic site that was opposite to the scissile thymine increased the enzymatic activity of RuvC and was able to rescue the activity defect of the R76A mutant. Our biochemical results were in good agreement with the results of our MD simulations.

**Simulations explain the DNA sequence recognition.** RuvC cleaves DNA at the 5′-A/TTT↓G/C-3′ consensus[5,11–13]. Our simulations offer an explanation for consensus recognition. They show that loss of the T–A base pair on the 5′ side of the scissile phosphate (i.e., the third nucleotide of the consensus) is required for catalytic geometry of the active site. This may explain the enzyme's ability to discriminate against G–C and C–G at this position because these base pairs are more stable, and their disruption would thus be less likely.

The simulations also explain why T–A is preferred over A–T on the 5′ side of the scissile phosphate. We often observed the formation of a new H-bond between the protein backbone and thymine O2 atom (Fig. 3). This H-bond would not form with adenine, which lacks an H-bond acceptor in this position. The adenine is also more bulky and could be a poor fit in this binding pocket. Finally, the simulations showed that disruption of the scissile T–A base pair can be connected with the formation of a stacking interaction between adenines flipped out from both cognate sequences and stabilized by the Arg76 side-chain (Supplementary Fig. 4). Such a stacking interaction would be

weaker with the thymine, which has a smaller base surface area. The Arg76 side-chain could be less effective in both disrupting the A–T base pair and maintaining the disrupted state than it is with the T–A base pair.

To test these assumptions, we conducted both standard and enhanced-sampling MD simulations, in which we inverted the base pair on the 5′ side of the scissile phosphate from T–A to A–T at both catalytic sites. In these simulations, the adenine of the A–T base pair did not form any new protein–DNA interactions. Furthermore, although spontaneous disruptions of the A–T base pair were observed, the Arg76 side-chain did not form such extensive interactions with the A–T base pair as it did with the T–A base pair, thus allowing eventual reconstitution of the base pair. Lastly, the cross-junction stack (Supplementary Fig. 4) was never observed in the REST2 simulations with the A–T scissile base pair, despite identical simulation timescales. This strongly contrasts with the REST2 simulations with the T–A scissile base pair, which revealed extensive cross-junction stacking of the adenines (Supplementary Notes). This difference may primarily result from the fact that the thymine base is smaller and cannot effectively cross the junction to form the cross-junction stack in the same fashion as adenine.

We also sought to understand the mechanism that underlies the recognition of other nucleotides in the cognate sequence, such as the last nucleotide that is either G or C. To explore this in a structural context, we performed MD simulations, in which we replaced the base pairs that were downstream of the scissile phosphate with T–A or A–T. In these simulations, conformational changes in the upstream scissile base pair often propagated to the succeeding (i.e., changed) base pair. When the upstream scissile base pair was lost, the downstream base pair was typically also disrupted (Supplementary Fig. 8), which was also accompanied by the loss of highly important interactions between the protein and DNA backbone that were further downstream (i.e., interactions that formed between the phosphodiester backbone and Ile10, Thr11, Lys83, and Arg47). Therefore, the preference for more stable G–C or C–G as the downstream base pair could be attributable to the fact that they prevent the propagation of disruptions along the HJ arm. Therefore, our findings explain the recognition of the second half of the consensus sequence. The mechanism of recognition of the second residue of the consensus may involve interactions with Phe73 or cross-junction stacking of adenines from the scissile pair and the second base pair of the consensus. Both possibilities are discussed further in the Supplementary Information (Supplementary Figs. 9 and 10).

**The nick-counternick mechanism relies on DNA relaxation**. An earlier study showed that the presence of partially cleaved HJ substrate accelerates its second cleavage by RuvC[14,15]. This observation implies structural communication between the two catalytic sites, in which the first DNA backbone cleavage alters the configuration of the second catalytic site, leading to faster cleavage.

To explore the structural basis of this mechanism, we performed MD simulations, in which we introduced a 5′-phosphorylated nick in the HJ DNA backbone at one catalytic site, corresponding to the product of the first RuvC enzymatic reaction. Our simulations showed that when the DNA backbone was nicked at the first active site, conformational changes in the scissile base pair that was located at the second catalytic site proceeded almost immediately after the start of the simulations. This contrasts with simulations of non-nicked DNA, in which the conformational changes occurred later during the simulations and were aided by the protein. This is likely because in the pre-nicked substrate the tension induced by the protein is relieved,

thus facilitating placement of the other scissile phosphate at the active site.

To verify the importance of the nucleotide sequence on the 5′ side of the scissile phosphate, we also performed a simulation in which we altered the scissile base pair at the second catalytic site into a non-cognate G–C base pair. In this case, no conformational changes in the scissile base pair were observed, even with the DNA backbone nick at the first catalytic site, likely because of the greater thermodynamic stability of the G–C base pair. This could suggest that even when the HJ is nicked, a cognate sequence is still required at the other active site.

To experimentally explore the mechanism of HJ cleavage coordination, we prepared a HJ substrate with a 5′-phosphory-lated nick at one active site (N-C). To examine the potential role of base flipping in the second HJ cleavage, we also prepared nicked substrates with abasic sites at a single cleavage site (N-Ab1A and N-Ab1B) or both cleavage sites (N-Ab2; Fig. 5a). Substrates with nicks 4 or 5 nt downstream from the expected cleavage site (N-C4, N-C5) served as controls. After verifying that the affinity of Tt-RuvC for both nicked and non-nicked substrates was similar (Supplementary Table 3), we performed activity assays that showed higher activity for nicked substrates, especially at early timepoints which is consistent with previous data[15]. Notably, the nicked substrates with abasic sites were cleaved only slightly more efficiently than the nicked substrates without any abasic sites (Fig. 5b, Supplementary Fig. 11). This suggests that substrate relaxation that is induced by the first nick alone is sufficient to accelerate the second catalytic event. We then performed activity assays using the R76A Tt-RuvC mutant that was markedly less efficient than the wild-type, although its activity was still greater with nicked than with non-nicked substrates (Fig. 5c; Supplementary Fig. 11). Only a nick located near or at the exchange point of DNA is expected to relax the structural tension and facilitate the second cleavage. In agreement with this, nicks located in the cleaved strands, 4 or 5 nt downstream from the expected cleavage site did not enhance the enzymatic activity of RuvC (Supplementary Fig. 12).

In summary, the nick-counternick mechanism appears to arise from the relaxation of tension around the HJ branching point upon the first cut that facilitates the necessary conformational changes near the second catalytic site.

## Discussion

We describe a mechanism that governs the sequence preference of RuvC and coordination of the two cuts that are introduced into HJ DNA (Fig. 6). We propose that both elements rely on similar structural determinants. Our higher-resolution RuvC–HJ crystal structure showed a distortion of base pair geometry around the HJ exchange point that was induced by the binding of RuvC (Fig. 1). At the same time, the scissile phosphates were positioned too far from the active sites for catalysis to occur, suggesting that tension that is introduced at the exchange point of the HJ substrate displaced scissile phosphates from the active sites (Fig. 6a, b). The MD simulations showed that this tension can be relieved by flipping out the adenine base that is opposite the thymine on the 5′ side of the scissile phosphate. This conformational change involves high-energy states that are not easily captured in the crystal structures but are actually responsible for positioning the scissile phosphate for the first cut (Fig. 6c). Notably, all of our crystallization trials in which we used HJs that contained a fully cognate sequence at the active sites of RuvC never yielded diffracting crystals, which could suggest a high level of disorder. Once the first cut is introduced, the tension is permanently released (Fig. 6d, e), allowing the second cut to proceed immediately and thus coordinating the two cleavage events. Our model

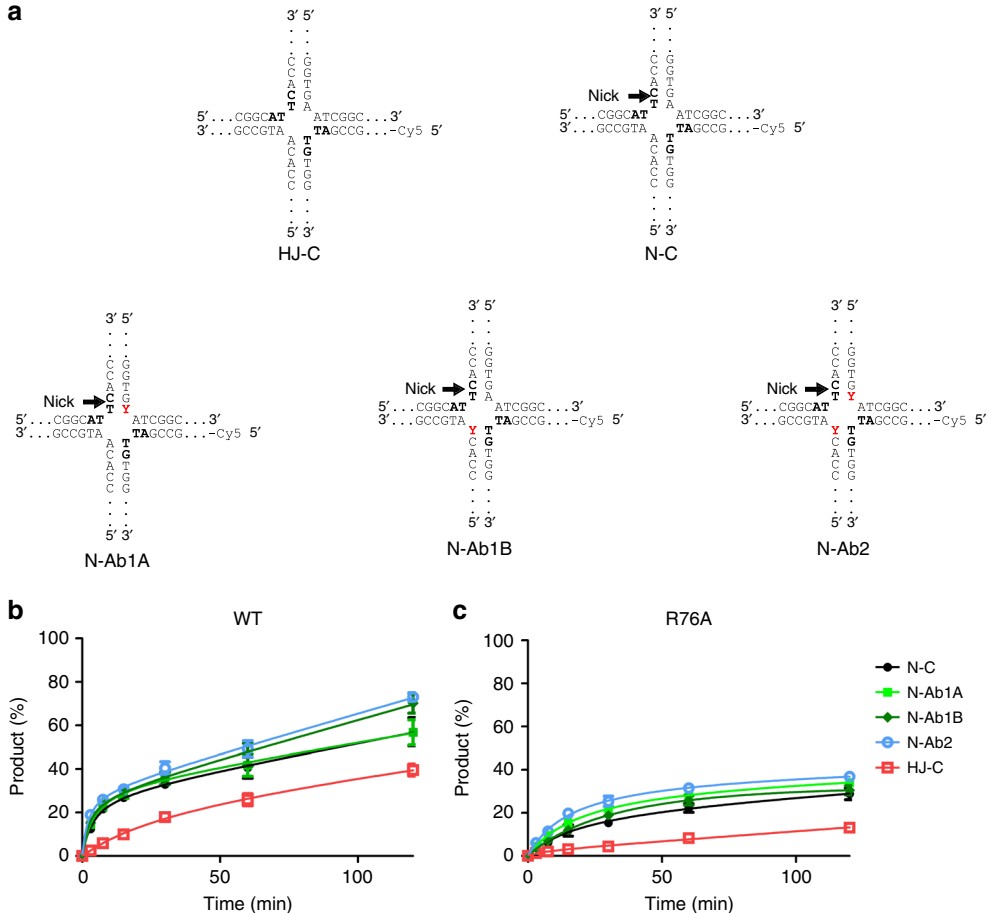

**Fig. 5** Activity of *Tt*-RuvC on nicked substrates. **a** Schemes of HJs that were used in the experiment. The red Y indicates abasic sites. **b**, **c** Resolving activity of the *Tt*-RuvC (**b**) and R76A (**c**) variant on the control and nicked substrates. Data from three independent experiments were averaged and plotted for each timepoint. Error bars represent the standard deviation

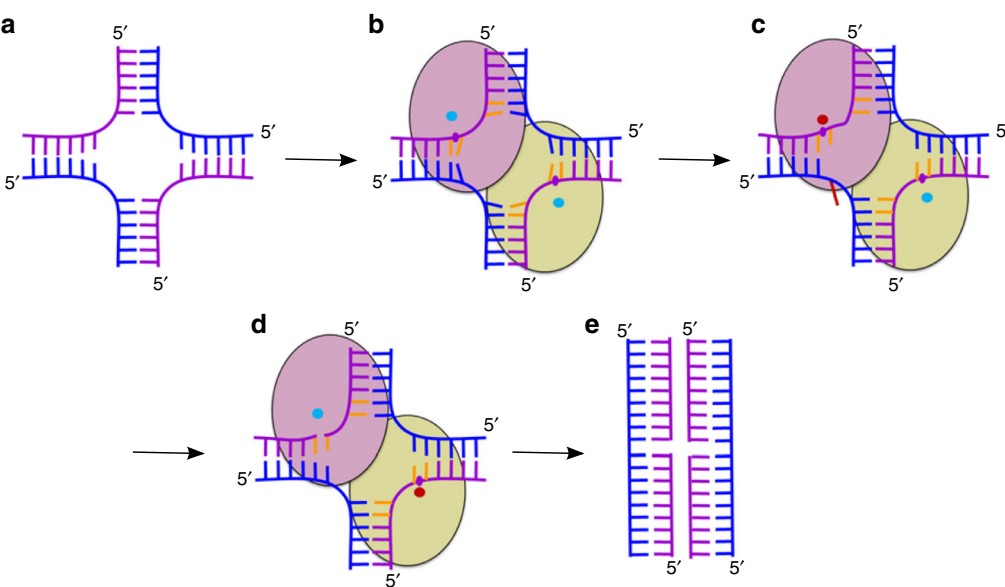

**Fig. 6** Cartoon representation of the mechanism of HJ resolution by RuvC. **a** Holliday junction. Cleaved and non-cleaved DNA strands are shown in purple and blue ladder-like representations, respectively. **b** Binding of the HJ DNA. The subunits of the dimer are shown as yellow green and pink ovals. The scissile phosphate is marked as a purple circle. Cyan circles show active sites in an inactive configuration. **c** Flipping of the adenine (red) opposite the scissile base. The active site in the catalytic configuration is shown as a red circle. **d** The second cut. **e** Resolution products

suggests that the same aspect of the RuvC–DNA complex structure (i.e., conformational tension around the exchange point and its release) allows the enzyme to both recognize the cognate sequence and execute the nick-counternick mechanism of cleavage. The mechanism we propose is in very good agreement with both experimental (structural and biochemical) and computational results. Thus, we would argue that it is very likely a dominant element of RuvC's enzymatic action. Nevertheless, the multi-pathway nature of the suggested mechanism does not rule out potential contributions of other factors. Additional details or alternative models could be obtained by future studies.

Ydc2 and Cce1 are yeast mitochondrial proteins that are closely related to RuvC. For *S. cerevisiae* Cce1, experiments with 2-aminopurine showed that the protein disrupts pairing for all four base pairs around the branching point of the HJ[26]. This confirmed earlier findings that showed unstacking of these base pairs upon protein binding[27]. Intriguingly, both Ydc2 and Cce1 also exhibit a sequence preference and cleave the HJ after a thymine residue. The wedge element is much larger in Cce1 (ref. [28]) and is likely to introduce larger disruptions at the exchange point of the HJ compared with RuvC. However, despite these differences, Ydc2/Cce1 could also utilize adenine flipping for consensus recognition and tension release at the exchange point to coordinate the cuts.

Holliday junction substrates were previously shown to exhibit significant conformational flexibility[29]. We performed a total of more than 100 µs of MD simulations of the RuvC–DNA complex and free HJs (Supplementary Notes, Supplementary Fig. 13). Our data suggest that resolvases, particularly RuvC, may have evolved to take advantage of the extensive dynamics of HJ substrates rather than entirely suppressing it upon binding. Recent single-molecule studies also revealed that the resolvase-DNA interactions are very dynamic[30]. Our study adds another layer to this complexity by showing a dynamic conformational readout of the DNA itself.

Our findings highlight an intriguing aspect of nucleic acid recognition. While dynamic changes of protein and nucleic acids during binding are well known[31,32], here we describe stochastic equilibrium dynamics occurring within context of a fully formed protein–DNA complex. This equilibrium dynamics is biologically significant since the ground energy state of the fully formed RuvC–DNA complex is not catalytically competent, and rare events leading to additional conformational changes are required for the reaction to occur. These changes are used to both probe the sequence of the substrate and concurrently coordinate cleavage at the two active sites. In other words, RuvC utilizes high-energy transient conformational states of the substrate to recognize the cognate sequence and to discriminate incorrect sequences in which the required high-energy states do not occur. We recently described another example of an indirect readout of the nucleic acid sequence by the protein through high-energy conformational states in the HIV-1 reverse transcriptase. In this enzyme, the conformational probing of the dynamic properties of the polypurine tract RNA/DNA hybrid and intrinsic potential for conformational changes from the chemically inactive ground state to the reactive rare conformational state are used to indirectly read the sequence[33].

We predict that additional proteins will be identified that utilize conformational changes for sampling of dynamic properties of RNA and DNA[23]. Studies of such mechanisms will require diverse methodologies, ranging from structural biology and computational methods to advanced biochemical approaches. The excellent agreement between the computational and experimental data in the present study provides a framework for performing similar studies of transient conformational states of nucleic acid enzymes and interdisciplinary computational and experimental studies in general.

## Methods

**Protein and Holliday junction preparation.** Protein preparation was performed as described previously[18]. Briefly, *T. thermophilus* RuvC (*Tt*-RuvC) expression plasmids were prepared based on the pET28 expression vector (Merck KGaA, Darmstadt, Germany). Wild-type and the R76A variant of *Tt*-RuvC proteins were expressed in the *E. coli* BL21 strain using induction with 0.4 mM isopropyl β-D-1-thiogalactopyranoside. Bacterial cells were resuspended in 40 mM $NaH_2PO_4$ (pH 7.0), 75 mM NaCl, 5% glycerol, and 1.4 mM β-mercaptoethanol, with the addition of a mix of protease inhibitors and lysozyme (final concentration of 1 µg/ml) and incubated on ice for 30 min. After sonication, imidazole was added to the cleared lysate to a final concentration of 10 mM and loaded onto a nickel column (GE Healthcare) that was equilibrated with 10 mM imidazole, 40 mM $NaH_2PO_4$, 500 mM NaCl, and 5% glycerol. The protein was eluted with a gradient of imidazole from 10 to 300 mM, and the fractions that contained the protein were dialyzed overnight in a buffer that contained 40 mM $NaH_2PO_4$, 75 mM NaCl, 5% glycerol, 0.1 mM dithiothreitol (DTT), and 0.5 mM ethylenediaminetetraacetic acid (EDTA). During dialysis, the tag was removed by PreScission protease cleavage, and protein was further purified on a Heparin column (GE Healthcare). The purified protein was eluted with a linear gradient of NaCl from 75 to 1000 mM. *Tt*-RuvC was stored in 20 mM HEPES (pH 7.0), 150 mM NaCl, 5% glycerol, 0.1 mM DTT, and 0.5 mM EDTA. For crystallization, the protein was concentrated to 16–26 mg/ml.

Unmodified oligonucleotides and oligonucleotides that were modified with Cy5, HEX, an abasic site, and 2-aminopurine were purchased from Metabion International AG (Martinsried, Germany). The sequences are shown in Supplementary Table 4. All of the HJs that were used in the biochemical assays were purified from native gel after annealing.

**Crystallization and structure solution.** For the crystallization experiments, *Tt*-RuvC (final concentration of 7 mg/ml) was mixed with the HJ at a 1.8:1 molar ratio, and EDTA was added to a final concentration of 5 mM. The complexes were mixed with the reservoir solution at an equal volume and crystallized by the sitting drop vapor diffusion method at 25 °C.

The crystals of the *Tt*-RuvC–DNA complex were obtained with oligonucleotides J221 and J222 (for sequences, see Supplementary Table 4). The crystals were grown in 0.4 M ammonium phosphate. They were large and regular but diffracted X-rays to only ~7 Å resolution. To improve X-ray diffraction, an experiment with controlled crystal dehydration using the HC1c-device at MX beamline 14.3 (Berliner Elektronenspeicherring-Gesellschaft für Synchrotronstrahlung [BESSY] II, Berlin, Germany)[34] was performed. Data were collected at room temperature with 93% humidity. The best X-ray diffraction dataset extended up to 3.4 Å resolution (Supplementary Table 1).

The structure was solved by molecular replacement with Phaser[35] using the previously described structure as a search model (Protein Data Bank [PDB] ID: 4LD0)[18]. The complete model of the complex was built in Coot and refined in phenix.refine (version 1.8.2–1309)[36] with rounds of manual building (Supplementary Table 1). The structure was refined to an $R_{free}$ value of 23.3% which places in the 90th percentile relative to all X-ray structures of similar resolution (3.3–3.5 Å). MolProbity score which combines the clashscore, rotamer, and Ramachandran evaluations is 1.9 (100th percentile, $N = 614$, resolution 3.409 Å ± 0.25 Å)[37]. The geometry is very good with only one Ramachandran outlier (the corresponding residue in the other protein chain of the RuvC dimer is in the allowed region and both adopt similar geometries). The structural model also has an excellent fit to the electron density maps. Only five protein residues have an real-space R-value Z-score (RSRZ) higher than 2 (percentile score relative to all X-ray structures of 77 and 68 for protein chains A and B, respectively). All DNA residues have RSRZ below 2. The higher-resolution structure allowed us to model side chains of amino-acid residues (Glu71, Gln72, Phe74, Tyr75, Arg76, and Trp86) that play roles in substrate recognition. The atomic coordinates of the *Tt*-RuvC–HJ complex were deposited in the Protein Data Bank (PDB ID: 6S16). The figures were prepared using Pymol (version 3.3.0, Schrodinger LLC).

**RuvC cleavage assay.** The cleavage assays were performed essentially as described previously[38]. The oligonucleotides that were used in the biochemical experiments are listed in Supplementary Table 4. They formed synthetic junctions with 25 base pair (bp) arms and fluorescently labeled cleaved strands that contained two cognate sequences: 5′-ATTC in the middle of one cleaved strand and 5′-ATTG in the other. The standard cleavage reaction mixture (10 µl) contained 500 nM RuvC and the substrate at a 2:1 molar ratio. The reaction buffer contained 20 mM bicine (pH 9.0), 100 mM NaCl, 1 mM DTT, 100 µg/ml bovine serum albumin, 5% glycerol, and 5 mM Mg acetate. The samples were incubated for 0–120 min at 60 °C, and samples were collected at the selected timepoints. For experiments with various substrate: protein ratios, to ensure protein solubility at higher concentrations, the reaction buffer was changed to 20 mM HEPES (pH 7.5), 150 mM NaCl, 1 mM DTT, 100 µg/ml bovine serum albumin, 5% glycerol, and 5 mM Mg acetate. The reactions were incubated for 30 min. The reaction was stopped by the addition of 10 µl of sample buffer that contained 95% formamide, 30 mM EDTA, and 0.1% bromophenol blue. The hydrolysis products were analyzed by 12% acrylamide gels with 20% formamide and 8 M urea. The reaction products were visualized with a

Typhoon Trio+ scanner (GE Healthcare), and the cleaved fraction of the substrate was quantified by densitometry.

**2-Aminopurine fluorescence measurements**. Buffer (50 µl) that contained 20 mM Tris (pH 8.0), 100 mM NaCl, 5% glycerol, 1 mM EDTA, 1 mM DTT, and 6 mM MgCl$_2$ was incubated for 10 min at 37 °C in the presence of the HJ. The protein was next added at a 2:1 molar ratio. Fluorescence emission spectra were obtained using a Jasco FP-8300 spectrofluorometer that was connected to a water bath to maintain a 37 °C temperature inside the cuvette. The excitation wavelength was set at 320 nm, and emission at 370 nm was recorded. The widths of both the excitation and emission slits were 5 nm. Fluorescence was measured using a quartz cuvette with a 10 mm path length. The results were corrected for background fluorescence by subtracting the spectrum of the buffer. We used HJ substrates with 25 bp arms. To prevent exchange point migration, no homology was present at the branch point. Only one of the strands contained the RuvC cognate sequence (5′-ATTG) at the exchange point.

**Measurements of protein-HJ binding**. The assays were conducted in black, 96-well, flat-bottom polystyrene NBS plates (Corning 3650) in a total reaction volume of 40 µl. The reaction buffer contained 20 mM bicine (pH 9.0), 100 mM NaCl, 1 mM EDTA, 1 mM DTT, 3% glycerol, and 10 mM CaCl$_2$. For the measurement of binding, protein in the reaction buffer was added to the plate wells to obtain final concentrations of 0.16, 0.31, 0.63, 1.25, 2.5, 5, and 10 µM. The final concentration of the Cy5-labeled HJ was fixed at 20 nM. The reactions were prepared in triplicate. The reactions were mixed by shaking for 5 s and incubated for 2 min at 25 °C. Immediately after incubation, fluorescence anisotropy was measured in a Tecan Infinite M1000 fluorescence microplate reader at an excitation wavelength of 635 nm and emission wavelength of 670 nm with a bandwidth of 5 nm. Binding curves for three independent series of measurements that were recorded for each system were analyzed globally by applying the appropriate three-parameter two-state model. All of the calculations were performed using Origin 2019 (www.origin.com).

**System building for molecular dynamics simulations**. We utilized the new X-ray structure of the RuvC–HJ complex (PDB ID: 6S16) as the starting structure in all of the molecular dynamics (MD) simulations. Molecular modeling was used to obtain structures with mutated amino acids, alternative DNA sequences, or nicked DNA substrates. The structure of free HJ DNA was obtained by removing the protein. The topologies and coordinates for the simulations were prepared in the tLeap module of AMBER 16[39]. The missing amino-acid side-chain atoms were automatically added by tLeap and visually inspected, and their positions were manually corrected. We extended the duplexes of HJ DNA where necessary so that each helical arm possessed at least 6 base pairs. We used ff12SB[40] and OL15[41] force fields to describe the protein and DNA, respectively. In all of the simulations, the simulated biomolecule was solvated in the octahedral box of SPC/E[42] water molecules with a minimal distance of 12 Å between the solute and the box border. The systems were neutralized by the addition of KCl ions, achieving an overall excess-salt concentration of ~0.15 M. In selected simulations, four Mg$^{2+}$ ions were directly placed at their expected binding positions near the non-bridging oxygen of the scissile phosphate within the RuvC catalytic sites[43]. In selected simulations, the positions of the Mg$^{2+}$ ions relative to the DNA and protein were further refined by distance restraints (see below). We used Joung[44] and Aqvist[45] parameters of KCl and Mg$^{2+}$ ions, respectively. Although the use of the latter parameters is not recommended for the description of bulk dynamics and the outer-shell binding of Mg$^{2+}$ ions[46], a recent study showed advantages of these older parameters when the inner-shell binding of Mg$^{2+}$ ions is modeled[47]. For a comprehensive justification of the utilized force-field parameters, see Supplementary Notes and[23].

**Molecular dynamics simulation protocol**. Standard equilibration and simulation protocols for protein-nucleic acid complexes were applied[48] (see Supplementary Notes for further details). We used the sander.MPI and pmemd.cuda modules of AMBER 16[39] to perform the equilibrations and production simulations, respectively. The SHAKE algorithm and hydrogen mass repartitioning were applied[49,50], allowing the use of a 4-fs-long integration step. A Berendsen thermostat and barostat[51] were used to regulate the temperature and pressure, respectively. In specific simulations (marked "rst" in Supplementary Table 2), a set of six flat-well distance restraints of selected pair-wise distances between atoms was used to establish catalytically relevant geometries within RuvC active sites (Supplementary Fig. 3). The goal of the distance restraints was to focus the simulations on investigating characteristic changes in the structure and dynamics of the complex that were induced by the DNA backbone interaction with the RuvC catalytic centers. The use of a relatively small number of simple distance restraints was deemed entirely sufficient for this purpose. Note that direct modeling of the RuvC enzymatic reaction was not the goal of the present study and would require a more thorough exploration of the free-energy surface of the catalytic center and the judicious application of quantum mechanical methods.

**Enhanced-sampling REST2 MD simulations**. We used Replica Exchange with Solute Tempering 2 (REST2)[52] enhanced-sampling simulations to further explore specific aspects of RuvC–HJ complex dynamics. In standard MD simulations, we observed early signs of possible extensive conformational changes in base pairing near the center of the DNA junction. Therefore, in our REST2 calculations, we included the four nucleotides in each arm of the HJ DNA that were closest to the branching point in the list of the atoms whose interactions with each other and the rest of the system were scaled (i.e., the so-called "hot region"). A total of 8 base pairs (16 nucleotides) were thus included in the hot region. In all of the REST2 simulations, interactions of the hot region atoms were scaled up to λ = 0.6. Eight replicas were used, achieving an overall average trajectory exchange rate of 25% between replicas. All of the REST2 simulations were performed under constant volume conditions, and a Langevin thermostat[39] was used to regulate the temperature. All of the other simulation settings were the same as in the standard MD simulations. All of the simulation trajectories were analyzed using the VMD[53] and cpptraj[54] programs. In the REST2 simulations, we analyzed both the individual replicas (discontinuous replicas, following the scaled Hamiltonian) and demuxed trajectories (continuous replicas, following the individual trajectories across the replica ladder)[23].

**Reporting summary**. Further information on research design is available in the Nature Research Reporting Summary linked to this article.

## Data availability

The authors declare that the data supporting the findings of this study are available within the paper its Supplementary Information and Data source files. Crystal structure data that support the findings of this study have been deposited in the Protein Data Bank with the 6S16 accession code. The MD simulation trajectories can be obtained from the corresponding author (MK) upon reasonable request.

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

## Acknowledgements

We thank the staff of beamline 14-3 at BESSY for assistance with data collection. This work was supported by the SYMBIT project (registration no. CZ.02.1.01/0.0/0.0/15_003/0000477), financed by the ERDF. This work was also supported by a Wellcome Trust International Senior Research Fellowship to M.N. (no. 098022). M.N. was a recipient of the Foundation for Polish Science Ideas for Poland award. The research was performed using the Centre for Preclinical Research and Technology (CePT) infrastructure (European Union project no. POIG.02.02.00-14-024/08-00).

## Author contributions

K.M.G. solved the crystal structure of the RuvC–HJ complex. K.M.G. and A.S. performed biochemical experiments. M.K. performed and interpreted MD simulations. J.P. analyzed the binding data. K.M.G., M.K., J.S. and M.N. wrote the manuscript.

## Additional information

**Competing interests:** The authors declare no competing interests.

