## [Peer Review File · Nature Communications]

Reviewers' comments:

Reviewer #1 (Remarks to the Author):

The manuscript by Górecka, et al., describes a comprehensive crystallographic, computational, and biochemical study on the binding, sequence recognition, and enzymatic activity of the classic Holliday junction resolvase RuvC. This enzyme binds four-way junctions at specific cognate sites. An earlier, lower resolution (3.8 Å) structure of RuvC with DNA showed how the enzyme bound to the junction in general, but did not show details that provide unequivocal mechanisms for how the enzyme recognizes the cognate sequence and catalyzes DNA cleavage. In this study the authors first present a nominally higher resolution structure (3.4 Å) of the protein-junction complex, providing evidence for a “wedge”, where amino acid side chains intervene into the DNA to induce buckling of the the base pairs. The crystal structure did not show a catalytic conformation, probably because the DNA sequence was only partially that of the cognate sequence, but serves as the starting point for a series of elegant, high-level molecular mechanics/dynamics (MM/MD) simulations to tease out the various molecular details for the mechanism of binding, specificity, and catalysis. A particular strength of the study is the validation of the predictions from the MM/MD simulations, using multiple approaches to monitor base pair opening and flipping and cleavage of variants of the enzyme and the DNA. The study presents strong evidence supporting a model in which the DNA must sample high-energy conformations (essentially providing conformational stress as an added force) that contributes to the concerted cuts that result in resolution of Holliday junctions. Thus, this comprehensive study would be of interest to a broad range of scientists interested in understanding mechanisms of DNA recognition and repair. The caveat is that the entire study is predicated on the reliability of the starting crystal structure, but the authors spend very little space in the main manuscript or the supplemental materials providing support for the important details of the structure (not that the structure is not believable, but at 3.4 Å, the amount of data available could lead to anomalies in the structure, and the authors simply have not provided details one would want).

Issues concerning the crystal structure that the authors need to address include the following:

1. The electron density map in Fig. S1 should be rendered as a stereo-view. There are details in the density map, in particular bulges extending to the left and right of the junction that may or may not trace the DNA backbone, or away from the backbone.
2. In addition, it is not clear whether the Fo-Fc map is a difference map of simply the refined structure minus the selected base pairs, or a true omit map (refined, selected residues removed, re-refined to erase memory of the phasing around these base pairs). The latter would be more appropriate, since with such a large structure, the refined phases will reproduce the density of the atoms that had been omitted.
3. Similar omit difference maps of the side chains of the wedge around these DNA base pairs, including the position of the R76 and F74 residues, and the important catalytic cations that were used in the simulations of the catalytic mechanism. The conformations of these residues are critical to the MM/MD simulations and, therefore, it is important that their starting positions are well defined with strong experimental evidence.

Reviewer #2 (Remarks to the Author):

Karolina Maria Górecka, Miroslav Krepl, Jarosław Poznański, Jiří Šponer and Marcin Nowotny determined the crystal structure of the complex of RuvC and Holliday junctions (HJs) at a better resolution (3.4 Å) than that reported previously by them (3.8 Å). RuvC is a canonical resolvase of HJs, which are four-way DNA structures formed as intermediates of homologous recombination. The questions that the authors addressed are how RuvC introduces two symmetric strand cuts at

specific DNA sequences, and how two cuts are tightly coordinated during the complete resolution of HJs by RuvC. Their higher resolution X-ray diffraction data revealed that they cause deformations of the base pairs that are closest to the junction point, and that the RuvC-HJ complex did not adopt a fully catalytic configuration. Starting from their new RuvC-HJ complex structure with the aid of molecular dynamic simulations and biochemical analysis, the authors claim that the correct positioning of the substrate for cleavage requires conformational changes within the bound DNA. They also claim that these changes involve rare high-energy states with protein-assisted base flipping that are readily accessible for the cognate DNA sequence but not for non-cognate sequences. In these studies, some features predicted from the molecular dynamic simulations were well supported by the biochemical analysis using mutant RuvC bearing a specific amino acid-replacement (R76A) and HJs with modified nucleotide or base-sequences. As for the coordinated two cuts which had been suggested by preceding biochemical studies by others, the authors performed MD simulations, and these results and their biochemical studies are consistent with the view that substrate relaxation that is induced by the first nick alone is sufficient to accelerate the second catalytic event. These findings are novel and stimulate our understanding about the resolution of HJs by RuvC.

So far presented, the results of the biochemical experiments are clear, although additional controls are required. Then, are their results sufficient to conclude that RuvC-HJ complex does adopt a fully catalytic configuration by the conformational changes within the bound DNA? It may not be required for the publication of their studies considering the current status of our understanding, but it is noted that the authors still do not directly showed by biochemical means or by structure analysis, the required structure for the sequence-specific cleavage induced by the conformational changes. This means that other possibilities for the molecular mechanisms of the recognition of the canonical sequence and coordinated two cuts in HJ by RuvC is not completely excluded.

The specific comments are listed below.

1. The authors showed biochemical activities of the wild-type or mutant RuvC with various substrates (Figs. 4a and 4d, and Fig. 5b) only at a single amount of protein and HJs in each experiment. The activities of an enzyme should be compared with varying the enzyme amount and/or the amounts of substrates, in authentic biochemistry.
2. The results shown in Fig. 4d and Fig. 5b lacked important and critical negative controls. As for Fig. 4d, the effects of the abasic site introduced some bases distant to the center of HJ should be tested, like the case of the experiments using 2-aminopurin shown in Fig. 4b. Likewise, in the experiments shown in Fig. 5b, the introduction of a nick at non canonical site should be tested. If these control experiments show positive results, the authors need to change their claims.
3. The authors described that they performed MD simulations, in which they replaced the base pairs that were downstream of the scissile phosphate with T-A or A-T. The authors should present the biochemical results using the same set of the HJ variants and mutant RuvC with the replacement of Phe73 to verify the results of MD simulations.
4. It should be specified which factors and parameters are considered in the molecular dynamic simulations that the authors performed. The authors described software names and cited references, but the general readers feel difficulty to understand how the authors performed the simulation. In addition, the movies of the simulation and the pymol files as supplementary information may help their understanding.
5. Lines 466-470, lines 474-486: The authors described that "in principle, specific interactions between nucleic acid enzymes and their substrates are conventionally considered contacts between rigid bodies where binding occurs through the complementarity of shapes and local interactions that then lead to the formation of a catalytically competent complex". This is not true. The dynamic structural changes have been considered since early 2000s or earlier, in the specific interactions between bio-macromolecules such as proteins and DNA. For example, an "intrinsically disordered scaffold" in protein-protein and protein-DNA interactions were published in 2005 (Mark et al. JMB), and an "intrinsically disordered protein" in 2007 (Sugase et al. Nature). Thus, the description "Our findings highlight an intriguing novel aspect of nucleic acid recognition" is invalid, and "The proposed mechanism of RuvC involves a different scenario" is not novel. The discussion relating to HIV-1 reverse transcriptase is no more significant.

The manuscript by Górecka, et al., describes a comprehensive crystallographic, computational, and biochemical study on the binding, sequence recognition, and enzymatic activity of the classic Holliday junction resolvase RuvC. This enzyme binds four-way junctions at specific cognate sites. An earlier, lower resolution (3.8 Å) structure of RuvC with DNA showed how the enzyme bound to the junction in general, but did not show details that provide unequivocal mechanisms for how the enzyme recognizes the cognate sequence and catalyzes DNA cleavage. In this study the authors first present a nominally higher resolution structure (3.4 Å) of the protein-junction complex, providing evidence for a “wedge”, where amino acid side chains intervene into the DNA to induce buckling of the the base pairs. The crystal structure did not show a catalytic conformation, probably because the DNA sequence was only partially that of the cognate sequence, but serves as the starting point for a series of elegant, high-level molecular mechanics/dynamics (MM/MD) simulations to tease out the various molecular details for the mechanism of binding, specificity, and catalysis. A particular strength of the study is the validation of the predictions from the MM/MD simulations, using multiple approaches to monitor base pair opening and flipping and cleavage of variants of the enzyme and the DNA. The study presents strong evidence supporting a model in which the DNA must sample high-energy conformations (essentially providing conformational stress as an added force) that contributes to the concerted cuts that result in resolution of Holliday junctions. Thus, this comprehensive study would be of interest to a broad range of scientists interested in understanding mechanisms of DNA recognition and repair. The caveat is that the entire study is predicated on the reliability of the starting crystal structure, but the authors spend very little space in the main manuscript or the supplemental materials providing support for the important details of the structure (not that the structure is not believable, but at 3.4 Å, the amount of data available could lead to anomalies in the structure, and the authors simply have not provided details one would want).

We would like to thank the referee for his/her remark that our study is comprehensive and of broad interest.

Based on referee’s suggestion, we provide additional data checks to verify the quality of our structure. We suggest that our structure is of high quality as judged by key parameters and their comparison with other structures in the PDB. Here is the graphical summary from the validation report with key parameters. Please note that they are all better than majority of the structures of similar resolution (white rectangles). Of note is a small R_{free} value – 23.3%, which puts the structure in the 90th percentile relative to all X-ray structures of similar resolution (3.3 -3.5Å).

[REDACTED]

The following information was added to the Methods section (page 17):

“The structure was refined to an R_{free} value of 23.3% which places in the 90th percentile relative to all X-ray structures of similar resolution (3.3 -3.5 Å). MolProbity score which combines the clashscore, rotamer, and Ramachandran evaluations is 1.9 (100th percentile, N=614, resolution 3.409Å ± 0.25Å).

The geometry is very good with only one Ramachandran outlier (the corresponding residue in the other protein chain of the RuvC dimer is in the allowed region and both adopt similar geometries). The structural model also has an excellent fit to the electron density maps. Only five protein residues have a real-space R-value Z-score (RSRZ) higher than 2 (percentile score relative to all X-ray structures of 77 and 68 for protein chains A and B, respectively). All DNA residues have RSRZ below 2." Short version of this information was added to Results on Page 5.

Importantly, the structure was very stable in MD dynamics simulations and this behavior was consistently observed in multiple plain MD simulations and enhanced-sampling simulations. This suggests that the force field was able to handle any potential inaccuracies arising from the lower resolution of our structure. In other words, its quality is entirely sufficient for the purpose of the computational study. We added a sentence on page 5 "The stable behavior of the structural model confirmed its sufficient quality and suitability for MD analysis". Importantly, we used other (higher resolution) structures from the PDB database showing the structurally related RNase H catalytic sites as a verification when we defined the catalytic-like geometries that we studied by our MD simulations of RuvC. This further limits the transfer of any potential inaccuracies into the computational data.

Issues concerning the crystal structure that the authors need to address include the following:

1. The electron density map in Fig. S1 should be rendered as a stereo-view. There are details in the density map, in particular bulges extending to the left and right of the junction that may or may not trace the DNA backbone, or away from the backbone.

We have prepared a stereo-view version of the Supplementary Figure 1. The bulges indeed correspond to phosphate groups. We chose a representation that demonstrates this more clearly.

2. In addition, it is not clear whether the Fo-Fc map is a difference map of simply the refined structure minus the selected base pairs, or a true omit map (refined, selected residues removed, re-refined to erase memory of the phasing around these base pairs). The latter would be more appropriate, since with such a large structure, the refined phases will reproduce the density of the atoms that had been omitted.

Yes, it is an omit map after refinement with simulated annealing to make sure we remove any model bias from the maps. This is now clearly stated in the legend of the Supplementary Figure 1.

3. Similar omit difference maps of the side chains of the wedge around these DNA base pairs, including the position of the R76 and F74 residues, and the important catalytic cations that were used in the simulations of the catalytic mechanism. The conformations of these residues are critical to the MM/MD simulations and, therefore, it is important that their starting positions are well defined with strong experimental evidence.

According to the reviewer's suggestion we calculated an omit map (after refinement with simulated annealing) for the wedge element (residues Phe74, Tyr75, Arg76). It is now presented in Supplementary Figure 1b. The catalytic metal ions were not observed in our structure. The positions of metal ions were modeled based on homologous structures as described on page 4 of the Supplementary Information.

Reviewer #2 (Remarks to the Author):

Karolina Maria Górecka, Miroslav Krepl, Jarosław Poznański, Jiří Šponer and Marcin Nowotny determined the crystal structure of the complex of RuvC and Holliday junctions (HJs) at a better resolution (3.4 Å) than that reported previously by them (3.8 Å). RuvC is a canonical resolvase of HJs, which are four-way DNA structures formed as intermediates of homologous recombination. The questions that the authors addressed are how RuvC introduces two symmetric strand cuts at specific DNA sequences, and how two cuts are tightly coordinated during the complete resolution of HJs by RuvC. Their higher resolution X-ray diffraction data revealed that they cause deformations of the base pairs that are closest to the junction point, and that the RuvC-HJ complex did not adopt a fully catalytic configuration. Starting from their new RuvC-HJ complex structure with the aid of molecular dynamic simulations and biochemical analysis, the authors claim that the correct positioning of the substrate for cleavage requires conformational changes within the bound DNA. They also claim that these changes involve rare high-energy states with protein-assisted base flipping that are readily accessible for the cognate DNA sequence but not for non-cognate sequences. In these studies, some features predicted from the molecular dynamic simulations were well supported by the biochemical analysis using mutant RuvC bearing a specific amino acid-replacement (R76A) and HJs with modified nucleotide or base-sequences. As for the coordinated two cuts which had been suggested by preceding biochemical studies by others, the authors performed MD simulations, and these results and their biochemical studies are consistent with the view that substrate relaxation that is induced by the first nick alone is sufficient to accelerate the second catalytic event. These findings are novel and stimulate our understanding about the resolution of HJs by RuvC. So far presented, the results of the biochemical experiments are clear, although additional controls are required. Then, are their results sufficient to conclude that RuvC-HJ complex does adopt a fully catalytic configuration by the conformational changes within the bound DNA? It may not be required for the publication of their studies considering the current status of our understanding, but it is noted that the authors still do not directly showed by biochemical means or by structure analysis, the required structure for the sequence-specific cleavage induced by the conformational changes. This means that other possibilities for the molecular mechanisms of the recognition of the canonical sequence and coordinated two cuts in HJ by RuvC is not completely excluded.

We would like to thank the reviewer for complimenting the novelty and clarity of our results.

We entirely agree that other explanations are not completely excluded. We added a text to the Discussion section on Page 14: “The mechanism we propose is in very good agreement with both experimental (structural and biochemical) and computational results. Thus, we would argue that it is very a dominant element of RuvC’s enzymatic action. Nevertheless, the multi-pathway nature of the suggested mechanism does not rule out potential contributions of other factors. Additional details or alternative models could be obtained by future studies.”

The specific comments are listed below.

1. The authors showed biochemical activities of the wild-type or mutant RuvC with various substrates (Figs. 4a and 4d, and Fig. 5b) only at a single amount of protein and HJs in each experiment. The activities of an enzyme should be compared with varying the enzyme amount and/or the amounts of substrates, in authentic biochemistry.

Indeed, points 1. and 2. identify important control experiments.

As suggested, we performed activity assays for wild-type protein and R76A variant on control HJ and double abasic substrate (Ab2) at 1:0.25 to 1:10 substrate:protein ratios. These data are now presented in new Supplementary Figure 7 (all subsequent Supplementary figures have been renumbered) and introduced on page 10 of the manuscript. These data confirmed the findings from the time-course experiments presented in the original version of the manuscript. For all substrate:protein ratios the wild-type protein had moderate activity on control HJ substrate which was greatly enhanced on abasic substrate. R76A had very low activity on a regular substrate and the activity of this mutant was rescued on abasic substrate. Thus, these results are in agreement with Arg76-assisted flipping of the adenine opposite the scissile thymine.

2. The results shown in Fig. 4d and Fig. 5b lacked important and critical negative controls. As for Fig. 4d, the effects of the abasic site introduced some bases distant to the center of HJ should be tested, like the case of the experiments using 2-aminopurine shown in Fig. 4b. Likewise, in the experiments shown in Fig. 5b, the introduction of a nick at non canonical site should be tested. If these control experiments show positive results, the authors need to change their claims.

We prepared additional control substrates with abasic sites in positions other than the flipped adenine: HJ-Ab1A_-4 had an abasic site in non-cleaved strand located 4 nucleotides upstream from the position of the adenine, HJ-Ab1A_5 had an abasic site 5 nucleotides downstream from the position of the adenine. These positions were further away from the exchange point than in 2-aminopurine substrate. This is because HJs used for activity assays have a homology region which enables migration of the exchange point, while the 2-aminopurine substrates were immobile.

The new substrates along with the regular HJ and the HJ with an abasic site replacing the flipped out adenine (HJ-Ab1A presented in the original version of the manuscript) were tested with wild-type protein and R76A variant. As expected, the new substrates did not enhance the enzymatic activity. These data are now described in the Results section on page 9 and presented in Supplementary Figure 6.

Similar to the control experiments with abasic site substrates, we prepared control substrates with nicks located in one of the cleaved strands and 4 or 5 nt downstream from the expected cleavage site. These sites were selected because based on the structure a nick in these positions should not perturb protein-DNA interactions. The activity of *Tt*-RuvC was tested on these substrates and, as expected, only a nick at the site of the expected cleavage enhanced the cut at the other consensus sequence. These data are now described in the Results section on page 13 and presented in Supplementary Figure 12.

3. The authors described that they performed MD simulations, in which they replaced the base pairs that were downstream of the scissile phosphate with T-A or A-T. The authors should present the biochemical results using the same set of the HJ variants and mutant RuvC with the replacement of Phe73 to verify the results of MD simulations.

We have prepared the F73A variant but in our reaction conditions, used for all our biochemical experiments, this variant was virtually inactive which precluded its further analysis. Only in the highest substrate:protein ratios a small amount of product was formed (please see figure below).

This result is in agreement with the studies of equivalent Phe69 in *E. coli* (Yoshikawa, Iwasaki et al. 2001) (PMID: 11152689) which showed that F69A and F69L variants were devoid of enzymatic activity.

This is now noted in Supplementary Information on page 7.

We need to note that F73A substitution in *T. thermophilus* has been tested before and small amount of enzymatic activity was observed (Chen, Shi et al. 2013) (PMID: 23118486). This differences between the results of Chen et al. compared with the published results for *E. coli* RuvC and our results for *Tt*-RuvC are likely attributable to different reaction conditions and substrates used.

4. It should be specified which factors and parameters are considered in the molecular dynamic simulations that the authors performed. The authors described software names and cited references, but the general readers feel difficulty to understand how the authors performed the simulation. In addition, the movies of the simulation and the pymol files as supplementary information may help their understanding.

We now include an extensive description and justification of all the utilized force field parameters as a new section entitled "Comments on the utilized force-field parameters and the MD simulation protocol" in the Supplementary Information (page 1). We also included a movie showing one of the MD simulations with Arg76-assisted adenine flipping. Pymol files are very large, but the MD simulation results are available upon request from one of the corresponding authors, Miroslav Krepl.

5. Lines 466-470, lines 474-486: The authors described that "in principle, specific interactions between nucleic acid enzymes and their substrates are conventionally considered contacts between rigid bodies where binding occurs through the complementarity of shapes and local interactions that then lead to the formation of a catalytically competent complex". This is not true. The dynamic structural changes have been considered since early 2000s or earlier, in the specific interactions between bio-macromolecules such as proteins and DNA. For example, an "intrinsically disordered scaffold" in protein-protein and protein-DNA interactions were published in 2005 (Mark et al. JMB), and an "intrinsically disordered protein" in 2007 (Sugase et al. Nature). Thus, the description "Our findings highlight an intriguing novel aspect of nucleic acid recognition" is invalid, and "The proposed mechanism of RuvC involves a different scenario" is not novel. The discussion relating to HIV-1 reverse transcriptase is no more significant.

This is a valid comment. We now refer to the two papers to stress that the dynamic behavior of protein-protein and protein-DNA interactions has been known. The intended message of our work is highlighting the importance of high-energy states and dynamic behavior for enzymatic activity on

DNA substrates. In accordance with this and referee's suggestions, we significantly rewrote this part of the Discussion (page 15) to read:

“Our findings highlight an intriguing aspect of nucleic acid recognition. While dynamic changes of protein and nucleic acids during binding are well known (Mark et al., 2005 JMB; Sugase et al., 2007, Nature), here we describe stochastic equilibrium dynamics occurring within context of a fully formed protein-DNA complex. This equilibrium dynamics is biologically significant since the ground energy state of the fully formed RuvC-DNA complex is not catalytically competent, and rare events leading to additional conformational changes are required for the reaction to occur. These changes are used to both probe the sequence of the substrate and concurrently coordinate cleavage at the two active sites. In other words, RuvC utilizes high-energy transient conformational states of the substrate to recognize the cognate sequence and to discriminate incorrect sequences in which the required high-energy states do not occur. We recently described another example of an indirect readout of the nucleic acid sequence by the protein through high-energy conformational states in the HIV-1 reverse transcriptase. In this enzyme, the conformational probing of the dynamic properties of the polypurine tract RNA/DNA hybrid and intrinsic potential for conformational changes from the chemically inactive ground state to the reactive rare conformational state are used to indirectly read the sequence (Figiel et al., 2018).”

REVIEWERS' COMMENTS:

Reviewer #1 (Remarks to the Author):

The authors have addressed the issues raised from the initial review, providing sufficient evidence that the structure, although still at modest resolution, is basically correct and supports the conclusions.

Reviewer #2 (Remarks to the Author):

Karolina Maria Górecka et al. revised their manuscript, in which the authors are properly responded all concerns that I had raised. Thus, I have no reason to hesitate in accepting this paper for publication. However, I found several points to be corrected or considered by the authors before publishing this paper.

1. Figure 1 (b and c), Figure 2 (b – d, at least, b and c), Figure 3 (at least, a and c), Supplementary Figure 1, Supplementary Figure 2, Supplementary Figure 4, Supplementary Figure 8, Supplementary Figure 9, Supplementary Figure 10, Supplementary Figure 13: It is very helpful for understanding the results and discussion, if the authors indicate in these figures, the polarity of the DNA strands with the labeling with "5' " and "3' " or arrowheads (5' to 3') with an explanation.

2. P. 5 line 96, "helices A1 and A2 are the cleaved arms; Figure 1b": It is advised that "helices" is replaced by "arms", since it is confused with "helices B" in line 684, line 695 and line 700 in p. 26. A1 and A2 are parts of DNA, and helices B appear to be helices of proteins.

3. Figure 1 (b): Are blue spheres attached to DNA strands the scissile phosphates? These spheres should be explained in the legend of this figure.

4. Figure 1 (b): The authors have to verify all labels "3' " and "5' ". Judging from the sequences J221 and J222 shown in Supplementary Table 4, the termini labeled with "3' " are likely to be 5' termini. In this figure, the DNA strand with the label "5' " of the arm B1 was shown to have 5' terminus in the arm A2 (the terminus opposite to the terminus labeled "3' "). The sequences J221 and J222 suggest that the arm B1 has no terminus like the arm B2, but that both of them have a terminal TTT loop or hairpin). If this is correct, the label "5' " and the apparent termini in the arm B1 are misleading, since these suggest that the arm B1 has strand termini, and thus, it is advised that the loop strand is indicated in the arm B1 or the author provide a description about this fact.

5. Line 3 of the legend for Supplementary Figure 2 (a): It is described that "The scissile phosphates are indicated with spheres.", but the spheres are missing on the DNA strands in this figure.

6. The labels of the X-axes of Supplementary Figure 7. "ratio Substrate : Product" should read "ratio Substrate : Protein".

7. Supplementary Figure 7. For experiments with varying substrate:protein ratios, the amounts of the substrates (HJ variants) seem to be fixed. Is it correct? The authors should provide the concentration of the substrates in molar concentration (or protein, if the protein amounts are fixed) in the legend of this figure.

Reviewer #1 (Remarks to the Author):

The authors have addressed the issues raised from the initial review, providing sufficient evidence that the structure, although still at modest resolution, is basically correct and supports the conclusions.

We are glad that we appropriately addressed the issues raised by this reviewer.

Reviewer #2 (Remarks to the Author):

Karolina Maria Górecka et al. revised their manuscript, in which the authors are properly responded all concerns that I had raised. Thus, I have no reason to hesitate in accepting this paper for publication. However, I found several points to be corrected or considered by the authors before publishing this paper.

We are glad that we addressed the issues raised by this reviewer, and we would like to thank him/her for very thoroughly reading of our manuscript.

1. Figure 1 (b and c), Figure 2 (b – d, at least, b and c), Figure 3 (at least, a and c), Supplementary Figure 1, Supplementary Figure 2, Supplementary Figure 4, Supplementary Figure 8, Supplementary Figure 9, Supplementary Figure 10, Supplementary Figure 13: It is very helpful for understanding the results and discussion, if the authors indicate in these figures, the polarity of the DNA strands with the labeling with “5’ ” and “3’ ” or arrowheads (5’ to 3’) with an explanation.

We introduced labels “5’ ” and “3’ ” or arrows indicating polarity of DNA strands to Figure 1, Figure 2, Figure 3 , Supplementary Figure 2, Supplementary Figure 4, Supplementary Figure 8, Supplementary Figure 10, Supplementary Figure 13.

2. P. 5 line 96, “helices A1 and A2 are the cleaved arms; Figure 1b”: It is advised that “helices” is replaced by “arms”, since it is confused with “helices B” in line 684, line 695 and line 700 in p. 26. A1 and A2 are parts of DNA, and helices B appear to be helices of proteins.

We checked and corrected this sentence.

3. Figure 1 (b): Are blue spheres attached to DNA strands the scissile phosphates? These spheres should be explained in the legend of this figure.

We checked and corrected legend in this figure.

4. Figure 1 (b): The authors have to verify all labels “3’ ” and “5’ ”. Judging from the sequences J221 and J222 shown in Supplementary Table 4, the termini labeled with “3’ ” are likely to be 5’ termini. In this figure, the DNA strand with the label “5’ ” of the arm B1 was shown to have 5’ terminus in the arm A2 (the terminus opposite to the terminus labeled “3’ ”). The sequences J221 and J222 suggest that the arm B1 has no terminus like the arm B2, but that both of them have a terminal TTT loop or hairpin). If this is correct, the label “5’ ” and the apparent termini in the arm B1 are misleading, since these suggest that the arm B1 has strand termini, and thus, it is advised that the loop strand is indicated in the arm B1 or the author provide a description about this fact.

We checked and corrected this figure. Indeed one label was misplaced, we removed it. We added short explanation to the legend of this figure about TTT loop in arm B1.

5. Line 3 of the legend for Supplementary Figure 2 (a): It is described that “The scissile phosphates are indicated with spheres.”, but the spheres are missing on the DNA strands in this figure.

We checked and corrected legend in this figure.

6. The labels of the X-axes of Supplementary Figure 7. “ratio Substrate : Product” should read “ratio Substrate : Protein”.

We corrected this figure.

7. Supplementary Figure 7. For experiments with varying substrate:protein ratios, the amounts of the substrates (HJ variants) seem to be fixed. Is it correct? The authors should provide the concentration of the substrates in molar concentration (or protein, if the protein amounts are fixed) in the legend of this figure.

We added information about concentration of the substrate in the legend of this figure.